# Selective Antimicrobial Therapies for Periodontitis: Win the “Battle and the War”

**DOI:** 10.3390/ijms22126459

**Published:** 2021-06-16

**Authors:** Mahmoud Elashiry, Ana Carolina Morandini, Celine Joyce Cornelius Timothius, Mira Ghaly, Christopher W. Cutler

**Affiliations:** Department of Periodontics, Dental College of Georgia, Augusta University, Augusta, GA 30912, USA; MELASHIRY@augusta.edu (M.E.); AMORANDINI@augusta.edu (A.C.M.); CCORNELIUSTIMOT@augusta.edu (C.J.C.T.); mghaly@augusta.edu (M.G.)

**Keywords:** periodontitis, antimicrobial therapy, antibiotics, antiseptics, NSAIDS, resolvins, antimicrobial peptides, NLRP3, inflammasomes, exosomes, dysbiosis, *Porphyromonas gingivalis*

## Abstract

Traditional antimicrobial therapies for periodontitis (PD) have long focused on non-selective and direct approaches. Professional cleaning of the subgingival biofilm by instrumentation of dental root surfaces, known as scaling and root planning (SRP), is the mainstay of periodontal therapy and is indisputably effective. Non-physical approaches used as adjuncts to SRP, such as chemical and biological agents, will be the focus of this review. In this regard, traditional agents such as oral antiseptics and antibiotics, delivered either locally or systemically, were briefly reviewed as a backdrop. While generally effective in winning the “battle” against PD in the short term, by reducing its signs and symptoms, patients receiving such therapies are more susceptible to recurrence of PD. Moreover, the long-term consequences of such therapies are still in question. In particular, concern about chronic use of systemic antibiotics and their influence on the oral and gut microbiota is warranted, considering antibiotic resistance plasmids, and potential transfer between oral and non-oral microbes. In the interest of winning the “battle and the war”, new more selective and targeted antimicrobials and biologics for PD are being studied. These are principally indirect, blocking pathways involved in bacterial colonization, nutrient acquisition, inflammation or cellular invasion without directly killing the pathogens. This review will focus on current and prospective antimicrobial therapies for PD, emphasizing therapies that act indirectly on the microbiota, with clearly defined cellular and molecular targets.

## 1. Introduction

Periodontitis (PD) is one of the most prevalent diseases globally, involving aging or inflammaging [1]. Once thought to be transmissible [2], there is no credible evidence for transmission of PD to an otherwise healthy host, though bacterial transmission does occur [3]. Thus, PD is not an infection in the traditional sense, as defined by Koch [4], but is a result of a polymicrobial consortium of key microbial species contained within the oral biofilm [5], called keystone pathogens [6]. These species exert undue influence on the commensal species and induce a dysbiosis, characterized by a disruption of the homeostatic balance needed to respond effectively, resulting in unregulated inflammation and alveolar bone loss. Traditional antimicrobial approaches to PD involve the direct killing of pathobionts, removal or disruption of the biofilm, with little to no selectivity. Here, we will discuss these modalities briefly, but will emphasize the newest indirect and selective approaches to achieve a more lasting therapeutic outcome.

## 2. Direct Antimicrobial Therapies

The advantages and disadvantages of conventional therapies that promote microbial clearance in and around the periodontal pockets are briefly reviewed to provide context for the new unconventional and pioneering therapies.

### 2.1. Systemically Administered Antibiotics

The use of antibiotics, taken perorally, combined with mechanical debridement, can help control periodontal pathogens in recalcitrant cases of periodontitis (PD). Both broad and narrow-spectrum antibiotics, alone or in combination, have been used as adjunctive therapy to SRP. Broad-spectrum antibiotics can eliminate or inhibit the growth of Gram-negative facultative and obligate anaerobes within the tissues. Typical examples include amoxicillin with or without clavulanic acid, azithromycin, ciprofloxacin, tetracycline, and doxycycline. Antibiotics with a narrower spectrum, such as metronidazole and clindamycin, are generally preferred with selectivity for anaerobic pathogens. Numerous interventional and observational studies highlight the clinical improvements observed in patients receiving systemic antibiotic therapy, particularly in association with mechanical periodontal therapy and oral hygiene instructions [7,8]. However, the question still remains as to whether these short-term gains are worth the cost in terms of antibiotic resistance and its effects on the oral and gut microbiome [9].

Systemic antimicrobials are principally indicated in immune-compromised PD patients or those with metabolic disorders that limit their response to mechanical therapy [10]. The ability of antibiotics to eliminate dysbiotic oral microbes, e.g., *Porphyromonas gingivalis,* from cellular reservoirs may be particularly relevant to efficacy [11,12]. Perhaps the most commonly prescribed antibiotic regimen for periodontal infections is a combination of metronidazole and amoxicillin [13,14,15]. The regimen ablates so-called ‘red complex’ pathogens [16] *Tannerella forsythia*, *Treponema denticola,* and *P.gingivalis,* and other potentially beneficial species [5]. Systematic reviews suggest that such a regimen may be useful, in the short term, in treating PD in otherwise healthy patients. Interestingly, this combination also significantly influences inflammation by increasing the stability of regulatory T cells (Treg), reducing their conversion to bone damaging Th17 cells [12]. Various randomized controlled trials have reported clinical benefits of adjunct systemic antibiotics in combination with non-surgical mechanical therapy; however, long-term benefits are inconclusive. Meta-analyses support the clinical benefits for PD, but these benefits decline after 12 months [17]. Antibiotics influence the community composition of the gut [18] and oral microbiomes [19]. Broad-spectrum antibiotics are contraindicated in patients with a history of *Clostridium difficile*-positive ulcerative colitis [20].

### 2.2. Locally Administered Antibiotics

Administering antibiotics locally can overcome many of the drawbacks of systemic delivery, precluding disturbances of the gut microbiome and patient compliance issues. Sustained release devices, including films, fibers, strips, gels, injectable devices, micro-and nanoparticles, have been used to treat PD since the introduction of tetracycline-infused fibers by Max Goodson in 1979 [21]. Sustained delivery in the periodontal pocket, subjacent to the site of bacterial invasion, makes intuitive sense, with increasing evidence to back it up. Systematic reviews report adjunctive reductions in probing depth, gingival inflammation, plaque scores, and bleeding indices when mechanical debridement was combined with local delivery drugs like minocycline hydrochloride (ARESTIN), chlorhexidine gluconate (PERIOCHIP), 10% doxycycline hyclate (ATRIDOX), and tetracycline hydrochloride (PERIODONTAL PLUS AB) compared to SRP alone [22]. Typical results with sustained local delivery devices in deep sites, following proper mechanical debridement and behavioral risk management, include reducing 0.4 mm in pocket depths and 0.3 mm gain in clinical attachment levels [23].

### 2.3. Antimicrobial Photodynamic Therapy (aPDT)

Although locally delivery of antibiotics is an effective alternative to systemic approach, the development of antibiotic-resistant strains has been reported [24]. Therefore, another approach for local delivery of antimicrobial agents such as aPDT has been investigated in clinical trials for periodontitis [25]. This technique is based on the use of a light-sensitive dye (photosensitizer) stimulated with visible light of an appropriate wavelength. Upon activation of the dye, free radicals of singlet oxygen are formed, cytotoxic to periodontal pathogens [26]. However, aPDT, when used as an adjunctive treatment, showed similar improvements in PD reduction and CAL gain compared with conventional periodontal therapy in periodontitis patients [25,27]. In addition, available evidence on the adjunctive therapy with aPDT is limited by the short-term follow-up, the low number of randomized controlled studies, and study design inconsistency [28,29].

### 2.4. Oral Antiseptics

Oral antiseptics have multiple applications in periodontal care, including home rinses, pre-procedural rinses, post-surgical care, and pocket irrigation. Particularly apt are applications for patients with physical or developmental disabilities, oral mucositis from radiation or chemotherapy, recurrent aphthous ulcers, and oral candidiasis. The most widely used classes of oral antiseptics are bisbiguanides (chlorhexidine), essential oils (eucalyptol, menthol, thymol), and quaternary ammonium compounds. These are often solubilized in alcohol and delivered via various vehicles: oral rinses, gels, pastes, chewing gums, lozenges, aerosols, varnishes, and sustained release devices (reviewed in Reference [30]). Mechanisms of action include permeabilization of the plasmatic membrane, precipitation of cytoplasmic proteins, cell wall disruption, inhibition of bacterial enzymes, extraction of endotoxins derived from lipopolysaccharide (LPS) of Gram-negative bacteria, and anti-inflammatory action based on antioxidant activity [31,32,33,34,35,36,37,38,39]. The chief advantage of chlorhexidine is its substantivity, or the ability to bind reversibly to oral tissues. This results in sustained antimicrobial effect (up to 12 h) [40,41,42].

Moreover, overgrowth of opportunistic microbes or microbial resistance over long-term use has not been observed [42,43,44], although other adverse effects are reported [45,46,47,48,49,50,51]. In a meta-analysis, Gunsolley [52] found chlorhexidine mouth rinses had the greatest reduction in plaque levels with the lowest heterogeneity among studies. In a systematic review with meta-analysis, Stoeken [53] reported that essential oil mouth rinses (e.g., Listerine^®^) significantly reduced both plaque and gingival indices but with significant heterogeneity between studies. Adverse side effects reported include burning sensation from the high alcohol content of certain formulations and tooth staining. Quaternary ammonium compounds (e.g., cetylpyridinium chloride [CPC]), in a meta-analysis, were found to confer significant benefits in plaque index and gingivitis index but with high heterogeneity among studies [52], with no changes in the oral microbiota or overgrowth of opportunistic species [53]. Side effects reported include tooth and tongue staining, transient gingival irritation, and aphthous ulcers [54]. Intrasulcular delivery of povidone-iodine, alone or in combination with antibiotics or chlorhexidine, has been shown to eliminate *P. gingivalis* infection commensurate with clinical improvement [55]. A twice-weekly oral rinse with 0.25% sodium hypochlorite produced marked decreases in dental plaque level and bleeding on probing [56]; however, long-term effects of this oxidizing agent on oral mucosa need to be assessed.

## 3. Indirect Antimicrobial Therapies

It is becoming increasingly clear that to win the battle and war against PD; novel antimicrobial strategies need to be developed that do not wipe out the resident flora. Instead, approaches targeting microbial mechanisms of colonization, nutrient acquisition, or accelerate their clearance by the immune system are required.

### 3.1. Probiotics

Probiotics are a well-accepted approach to treating dysbiotic conditions of the gut, such as *Clostridium difficile* infection. Fecal microbiota transplants from healthy patients can restore microbial homeostasis in the gut by replacing pathogens with commensal microbiota [20]. However, the application of probiotics for gingivitis and PD is presently in its infancy, with long-term studies yet to be carried out. Nonetheless, intriguing studies have been reported (reviewed in [57]. For example, *Lactobacillus reuteri* was applied in a randomized, placebo-controlled clinical trial to a small cohort of 59 gingivitis patients over two weeks. The results indicate significant improvement in plaque index and gingival index in the test group, not observed in the placebo group [58]. Recent studies also showed that an adjunctive consumption of *Lactobacillus* reuteri lozenges improved CAL change at molar sites with ≥5 mm deep pockets [59] without influencing pocket colonization with periodontal pathogens [60]. Another meta-analysis supported the adjunctive use of L. reuteri to SRP in deep pockets [61]. In an 8-week study, *Lactobacillus salivarius* WB21 was applied, in tablet form, to volunteers without severe PD, using a randomized, double-blind, placebo-controlled study design. Significant improvements in plaque index and probing pocket depth was observed in current smokers in the test group [62]. The use of a probiotic yogurt supplemented with *Bifidobacterium animalis* was shown to have a positive effect on reducing plaque accumulation and gingival inflammatory parameters [63]. Another study suggested that the use of *Bifidobacterium lactis* HN019 as an adjunct to SRP could provide additional clinical, microbiological, and immunological improvements in periodontitis patients [64]. Further long-term studies in larger patient cohorts and that measure clinical attachment loss and alveolar bone are required before conclusions can be drawn about clinical efficacy of oral probiotics.

### 3.2. Host Modulation with NSAIDS, Bisphosphonates, and SDD

Uncontrolled inflammation, triggered by microbial dysbiosis [6], is the principal cause of tissue destruction in PD. Dysbiotic oral pathogens benefit from inflammation and tissue proteolysis products, essential nutrients for their growth [65]. Thus, a vicious cycle occurs, with infection triggering inflammation and inflammation fueling infection. Therefore, a prudent strategy is to break the cycle by targeting the inflammation, infection, or both. This can consist of selectively starving the pathogens, preventing their colonization, enhancing their clearance [66], or resolving inflammation [67]. Host modulation using agents such as non-steroidal anti-inflammatory drugs (NSAIDS), bisphosphonates, and tetracycline have been reported. The sole drug approved by the FDA in this regard is sub-antimicrobial dose doxycycline (SDD). In early studies by Golub, short-term administration of SDD reduced collagenase activity [68]. Subsequent clinical trials increased the dosing regimen to longer periods, showing a reduction of inflammatory biomarkers. Cessation of therapy reversed collagenase activity to pre-treatment levels [69]. The increased duration of SDD therapy to regimens of 3 and 6 months up to 1 and 2 years have been proven with significant clinical efficacy, safety, and substantivity by various types of studies that followed [70,71,72,73,74,75]. These types of therapies inhibit collagenases without applying selective pressure that could drive antibiotic resistance.

### 3.3. Monoclonal Antibodies/Cytokine Inhibitors

More selective and targeted strategies are needed to modulate collateral tissue damage. Cytokine inhibitors such as anti-TNF-α, anti-IL-17A, and anti-IL-6 have been used in other inflammatory-driven diseases such as rheumatoid arthritis (RA). Compelling evidence shows that circulating levels of TNF-α are associated with the more severe manifestation of PD, especially in the elderly population [76]. The influence of treatment of rheumatoid arthritis with infliximab (anti-TNF-α) on comorbid PD has been examined [77]. Intriguingly anti-TNF therapy unexpectedly aggravated gingival inflammation. However, attachment loss was decreased [77], with the overall conclusions drawn that periodontal tissue destruction and gingival inflammation are distinct events in PD pathogenesis. Comparative studies of patients with autoimmune diseases, including rheumatoid arthritis, demonstrated that these patients have higher periodontal indices (bleeding on probing, pocket depth, clinical attachment loss) and higher TNF-α levels in gingival crevicular fluid than healthy controls, with anti-TNF-α effectively reversing this disparity [78,79]. Another study compared the periodontal condition in patients with RA and PD before and after therapy with an IL-6 receptor inhibitor, tocilizumab. Anti-IL-6 receptor therapy showed beneficial effects in serum inflammatory mediators, decreasing serum levels of TNF-α, total IgG, and serum amyloid A, though serum IL-6 and soluble IL-6R were significantly increased [80]. Neither study analyzed the influence on alveolar bone loss. In diabetic mice subjected to *P. gingivalis* oral infection, anti-TNF-α antibody reduced serum TNF-α, IL-6, and fasting blood glucose levels [81]. This same study showed marked improvement of wound healing in diabetic mice after *P. gingivalis* inoculation [81]. Wistar rat study of ligature-induced PD showed infliximab significantly reduced granulocyte blood counts, gingival IL-1β, TNF- α, and MPO levels, and diminished MMP-1/-8 RANK, and RANK-L in bone. Periodontal histopathological scores were also improved, and alveolar bone loss was inhibited [82].

### 3.4. Pro-Resolving Mediators

In this context, the ‘keystone pathogen’ hypothesis of PD pathogenesis [6] should be reiterated. Keystone pathogens, such as *P. gingivalis*, are those that assert undue influence on the local microbiome, causing dysbiosis and resulting in chronic inflammation. In parallel, but not at odds with this hypothesis [83], is the notion that unresolved inflammation, especially neutrophil-mediated, is the principal driving force for pathogenesis (reviewed in [84]). This notion has received considerable attention of late, involving novel therapeutic approaches targeting neutrophil emigration and clearance through manipulation of CXCL8 [85,86,87] or applying pre-resolving mediators. The latter include resolvins, protectins, maresins, and lipoxins. Given the importance of leukocyte trafficking in inflammation, endogenous positive and negative signaling mediators of inflammation have been referred to as local ‘go’ and ‘stop’ signals. Lipoxins (LX) are an important ‘stop’ signal produced in vivo during inflammation [88]. In an experimental PD model in rabbits, application of LXA4 or over-expression of 15-lipoxygenase promoted reduced inflammatory phenotype and were protective against alveolar bone loss [89]. Resolvin E1 (RvE1) has emerged as particularly efficacious. RvE1 is biosynthesized from eicosapentaenoic acid (EPA) and selectively interacts with specific receptors to inhibit leukocyte infiltration, obtund cytokine generation, and promote PMN apoptosis. The latter favors PMN clearance by macrophages and restoration of tissue homeostasis [90]. Nanomolar doses of RvE1 inhibit RANKL-induced osteoclast growth and differentiation, downregulating bone resorption in vitro [91]. A study of human periodontal ligament stem cells showed that in pro-inflammatory milieu, pluripotency, viability, and cell migration were suppressed, whereas maresin-1 (MaR1) and RvE1 restored tissue regenerative capacity [92]. RvE1 application is also effective at promoting bone preservation in the mouse calvaria model [93] and regenerating bone in the ligature-induced periodontitis in rats [94]. Moreover, shifts in the subgingival microbiota, i.e., dysbiosis, induced with ligature were markedly altered by RvE1; namely, *P.gingivalis* was reduced. Simply put, RvE1-mediated regulation of inflammation appears to reverse the dysbiosis coincident with prevention and treatment of inflammatory disease [94].

Regarding the clinical relevance of the pro-resolving mediators, a recent cross-sectional case-control study investigated whether salivary levels of LXA4, protectin D1 (PD1), RvE1, and MaR1 might be predictive of periodontal health versus disease status [95]. Compared with healthy controls, detected in PD patients was a significant decrease in LXA4, an increase in PD1/MaR1, but no difference in RvE1 salivary levels. In addition, clinical parameters such as probing depth and clinical attachment loss were negatively correlated with LXA4, positively correlated with PD1/MaR1, and not correlated with RvE1 salivary levels, suggesting an imbalance of pro-resolving lipid mediators in patients with PD [95]. There have been no published reports of clinical efficacy of pre-resolving mediators for PD in humans.

### 3.5. Biologics

Here, we briefly review peptides derived from the innate immune defense system (anti-microbial peptides), from oral microbes, and specific inhibitors of NLRP3 inflammasome or senescence (senolytic agents).

#### 3.5.1. Antimicrobial Peptides

Antimicrobial peptides are a diverse and versatile class of small molecules derived from the innate immune system, with particular efficacy for treating polymicrobial biofilm infections (reviewed in [96]). These generally have a broader spectrum than antibiotics, promote wound healing, and have a low probability of resistance [96]. Potential limitations include susceptibility to protease degradation, sequestration by biological fluids, inactivation by physiological concentrations of salts, and potential toxicity towards eukaryotic cells [96]. These factors complicate the regulatory approval process, delaying the immediate application of antimicrobial peptide-based therapeutics in oral polymicrobial infections. Mention should be made of the synthetic cationic antimicrobial peptide, Nal-P-113. It has shown significantly higher antimicrobial activities than penicillin, chlorhexidine, and metronidazole against Gram-positive *Streptococcus gordonii* Challis CH1, Gram-negative *Fusobacterium. nucleatum* ATCC25586 and *P. gingivalis* W83 [97].

#### 3.5.2. Targeted Microbial Peptides

Fimbriae are important virulence factors used by *P. gingivalis* during colonization, as they mediate adhesion to other oral bacteria and host cells. This makes fimbriae an attractive candidate for antimicrobial therapies targeted to treat PD. The colonization of the subgingival pocket by *P. gingivalis* occurs through multispecies interactions of early and late colonizers. Among the species that are reported to successfully facilitate *P. gingivalis* adherence to the dental biofilm are *F. nucleatum* and commensal streptococci species such as *S. gordonii*, and the initial interactions between them represent a viable target for therapeutic intervention to limit periodontal pathogen colonization. Early studies from Lamont RJ et al. demonstrated that a protein–protein interaction mediates adherence of P. *gingivalis* to streptococci between the minor fimbrial antigen (Mfa) and streptococcal antigen I/II (e.g., SspB) [98,99]. They also demonstrated the in vivo effectiveness of a peptide inhibitor of *P. gingivalis* adherence in reducing bone loss in a dual-species biofilm model of *P. gingivalis-S.gordonii*-infected mice [100]. More recently, the same group also identified a peptidic inhibitor derived from *Streptococccus cristatus* that represses the expression of virulence factors of *P. gingivalis,* including fimbrial proteins and gingipains [101,102]. They also showed that the streptococcal derived anti-*P gingivalis* peptide, called SAPP, was able to reduce the levels of other Gram-negative bacteria strongly associated with periodontitis, such as *Tannerella forsythia*, *Treponema denticola,* and *F. nucleatum* [103]. Another recent report showed that peptides derived from conserved C-terminal regions of the Mfa1 and FimA subunits inhibit fimbrial assembly on the surface of *P. gingivalis* and interfere with fimbrial function in biofilm formation [104]. These are encouraging results that need to be further examined in PD. These findings lay the groundwork for the development of antiinfective drugs targeting specific proteins associated with the adherence of a “keystone” pathogen and major player in establishing a dysbiotic polymicrobial periodontal environment [105]. Fimbriae are also one of the virulence factors that allow *P. gingivalis* to escape the immune system strategies developed to fight the infection. It has been shown that *P. gingivalis* can dampen IL-1β secretion in murine macrophages by means of its major fimbriae compared to the wild-type strain [106].

#### 3.5.3. Inhibitors of NLRP3 Inflammasome

The induction of inflammatory processes in the host cells often requires the engagement of inflammasomes, which are protein platforms that aggregate in the cytosol in response to different stimuli such as microbial infection or cell damage. Among the sensors of inflammasomes, the nucleotide-binding domain leucine-rich repeat-containing proteins (NLR), of which NLRP3 (pyrin domain containing 3) is the most studied. Activation of NLRP3 results in a caspase 1-dependent release of cytokines IL-1β and IL-18 and lytic programmed cell death called pyroptosis [107,108]. A distinctive pathologic feature of PD is the excessive production of IL-1β [109]. Moreover, inflammatory responses against *P. gingivalis* infection are dominated by IL-1β. Expression of NLRP3 is higher in PD patients and significantly downregulated by conventional periodontal treatment [110]. Therapeutic targeting of NLRP3, most notably by small molecules such as the diarylsulfonylurea compound MCC950, has received enormous attention of late. MCC950 is particularly potent in inhibiting canonical and non-canonical NLRP3 inflammasome activation in mouse and human macrophages [111]. MCC950 has demonstrated therapeutic efficacy in various preclinical immunopathological settings [107], most notably in experimental PD in mice [112]. MCC950 significantly suppresses alveolar bone loss and caspase-1 activation in aged mice. Moreover, IL-1β-mediated RANKL-induced osteoclastogenesis due to macrophages was suppressed in vitro by MCC950 [112].

#### 3.5.4. Complement C3 Inhibitors

The complement system plays an important role in innate immune surveillance through its cascade of proteins, convertases, and receptors that bind inflammatory cells. Complement is activated by pathogen-associated and danger-associated molecular patterns (PAMPs, DAMPS, respectively). Activation of complement culminates in complement C3 activation [113]. Pathological C3 activation is implicated in many chronic diseases, including periodontitis [114]. Mice deficient in C3 or C3aR are resistant to gingival inflammation and bone loss [115,116]. Classes of therapeutic peptides have been developed to target complement C3, most notably Cp40, a cyclic peptidic inhibitor of C3, termed AMY-101 [117]. AMY-101 has completed phase 1 safety trials in humans [117] and has received Investigational New Drug approval by the U.S. Food and Drug Administration for the first clinical trial for gingival inflammation.

## 4. Exosome-Based Therapies

The discovery in 1987 of exosomes (EXO) [118] was a major breakthrough in understanding how cells communicate with other cells (reviewed in Reference [119]). EXO are nano-sized extracellular vesicles secreted by all cells. Originally considered cellular waste, EXO have emerged as important ‘packets’ of molecular information, reflective of the physiologic and pathologic state of the source cell [120,121,122,123,124]. Their presence in most body fluids and content of a repertoire of proteins [124,125,126,127], mRNAs, and miRNAs [124,128] has raised excitement for potential in disease diagnostics [129]. Saliva exosomes, for example, are under intensive study for early disease biomarkers [130]. Exogenously created EXO are being used for various therapeutic applications [131,132]. These include the use of dendritic cell (DC) derived immunoregulatory EXO for experimental PD [133]. Distinguishing features of EXO include their size (20–150 mm), shape [134], gradient density [135], and mode of biogenesis. The extracellular domain of EXO contains various adhesion molecules, including tetraspanins and integrins that promote binding to and activation of host immune cells [132,133,136] and regulation of antigen-presenting activities [132]. The miRNA content [133] and proteome [136] of dendritic cell-derived exosomes, both natural and engineered to regulate alveolar bone loss, have been extensively characterized by our group. EXO protects their cargo from proteolytic degradation [133] and can transfer it to local acceptor immune cells in the gingiva, for example [133], or to distant sites through the bloodstream, such as the lungs, liver, and secondary lymphoid organs [136]. Several other studies have established the capability of EXO as a natural nano delivery approach for a variety of infectious and immune/inflammatory diseases [137,138,139,140,141,142,143].

### 4.1. Endogenous EXO: Infectious and Inflammatory Diseases

The “dark side” of endogenous EXO has also been recognized. EXO from human immunodeficiency virus (HIV) infected cells carry the viral proteins and induce apoptosis of uninfected CD4 T cells and immune suppression [144,145,146,147,148]. Serum EXO from HCV-infected patients contains HCV RNA and induces HCV transmission to liver cells through viral receptor-independent mechanisms [149]. The plasma of mice with transmissible spongiform encephalopathy (TSE) contain EXO that harbor infectious misfolded prion proteins responsible for brain damage [150]. EXO also have a role in the bacteria–host interaction. EXO derived from *P.gingivalis*-infected DCs are enriched in MFA-1 fimbrial proteins, involved in binding to, infecting, and regulating the functions of bystander immune cells [151]. Serum EXO isolated from patients with tuberculosis infection was found to carry mycobacterial antigens [152]. 

Moreover, *Myobacterium tuberculosis* or *Myobacterium bovis*-infected macrophages secrete EXO-bearing bacterial antigens to stimulate antigen-specific immune responses [153,154]. Overall these studies suggest a pathologic role for EXO in disseminating infection to recipient cells [150]. EXO have also been implicated in the pathogenesis of inflammatory disorders and autoimmune diseases. For example, EXO of intestinal lumen aspirates of inflammatory bowel disease IBD patients carries pro-inflammatory factors including TNF-α, IL6, and IL8 in levels higher than that of healthy controls. These EXO activate epithelial cells and macrophages to induce IL-8 secretion [155]. EXO purified from synovial fibroblast of rheumatoid arthritis patients contain TNF-α at the transmembrane domain that induce NF-κB activation and MMP-1 release in acceptor / by stander cells [156]. Furthermore, EXO secreted from atherogenic macrophages were found to promote atherosclerosis. This could be through the transfer of specific EXO miRNAs to recipient macrophages, inhibiting migration and promoting their entrapment in the blood vessel wall [157].

### 4.2. Exogenous EXO: Antiviral, Antibacterial, and Anti-Inflammatory Nano-Delivery Systems

EXO has several advantages as therapeutic nano-delivery systems, including relative ease of storage, phenotypic stability, and in vivo delivery. EXO can be engineered to target specific pathologic pathways [132,158,159,160,161,162,163,164,165]. Our recent work has shown that DC EXO, engineered with IL-10 and TGFb, target STAT3 and SMAD in DCs and T cells [136] and reprogram inflammation towards a bone-sparing Treg response [133]. The maturation stage of the source DCs can determine whether EXO are immunostimulatory or immunosuppressive. Mature DC EXO (mDC EXO), loaded with specific peptides, can elicit potent specific immune activation, resulting in tumor cell eradication or microbial elimination [142,166,167]. In clinical trials, cancer patients vaccinated with autologous mDC EXO showed additional anti-cancer effects without significant side effects [168,169]. EXO from DCs pulsed with bacterial antigen peptides expresses specific bacterial proteins that can be utilized to develop prophylactic/therapeutic bacterial cell-free vaccines [154,170]. Furthermore, DCs pulsed with *Toxoplasma gondii* secrete EXO that promotes anti-parasite immunity in mice [171]. On the contrary, immature DC EXO (imDC EXO) can be modified to induce immune tolerance in immune disorders and animal models of transplantation. Immuno-suppressive modifiers of EXO, including IL-10, IL4, TGFB, FasL, IDO and CTLA4, can obtund inflammatory immune disorders [158,159,161,162,163,164,165,172]. ImDCs EXO inhibits T cells activation and proliferation directly or indirectly by reprograming the biological function of acceptor DCs and promoting a regulatory/ suppressive subset [162,164,172]. Previous reports demonstrated the association of imDCs EXO with acceptor CD11c+ cells in the dermis and regional lymph nodes after intradermal injection in an inflammatory disease model. These EXO were also found to be internalized by macrophages and CD11c+ DCs in the liver and spleen when delivered systemically through the intravenous route [162]. imDC EXO carrying suppressive factors were shown to inhibit rheumatoid arthritis and prevent the progression of joint destruction in mice [161,162,163,164]. In an animal model of inflammatory bowel disease (IBD), TGF-β1 containing imDCs EXO abrogated the disease severity and clinical manifestations while inhibiting Th17 responses and upregulating T-regulatory cells (Tregs) [172]. imDCs EXO with membrane-bound TGF-β1 have also been shown to inhibit experimental autoimmune encephalomyelitis in mice [173]. In another study, imDC EXO was found to prolong cardiac and liver transplantation [174,175]. Our group previously studied the immunobiology of DC-derived EXO subtypes isolated from murine DCs at different maturation stages. The most noted observation was the potency of immune-regulatory EXO (regDCs EXO) subsets, loaded with TGFb1 and IL10, in reprograming the immune function of acceptor DCs and CD4+ T cells towards a regulatory response in vivo. In addition, these EXO were retained at the inflammatory sites and protected their therapeutic cargo thus abrogating inflammatory bone loss in periodontitis mice models [133]. We also investigated in-depth the proteomic cargo and biodistribution characteristics of these immune therapeutic EXO. RegDC EXO were found to contain a complex protein cargo profile [136], enriched in proteins related to immunoregulation, trafficking to tissue, and cell binding. Moreover, regDCs EXO showed preferential biodistribution and accumulation into lung tissue when injected through the tail vein in mice. In addition, regDCs EXO were shown to inhibit the SARS-CoV-2 target receptor, ACE2, expression in a TGFb1 dependent manner in respiratory tract epithelial cells (PBTECs). These observations suggest immunotherapeutic implications of regDC EXO for COVID-19 infection. Reprogramming the key immune cells in the lungs, including DCs and T cells, toward a regulatory phenotype could reverse harmful inflammatory responses, as reported [133]. In addition, blocking the entry point of SARS-CoV-2 with regDC EXO treatment to attenuate the severity of the infection could be another mechanism [176]. EXO could also be engineered to express decoy ACE2 [177] or the S protein of the SARS-CoV-2 to be used as a vaccine [178]. In addition, EXO loaded with antiviral drugs is another strategy in fighting COVID19 infection [179]. EXO have been isolated from other immune cells to tap into their therapeutic potential, including Tregs [180,181,182], B lymphoblasts [183,184], natural killer cells [185], and mast cells [186]. Notably, EXO isolated from macrophages have been loaded with antibiotics, including linezolid and vancomycin, to inhibit intracellular infection by methicillin-resistant *Staphylococcus aureus (MRSA)* [187,188]. EXO derived from muscle cells loaded with viral antigens to be used as a vaccine and induce specific cytotoxic T lymphocyte immunity [189]. In summary, EXO appear to be a good substitute for whole cell-based therapy in use as anti-inflammatory agents and antimicrobial subcellular vaccines [132,158,159,160,161,162,163,164,165]. This could be a promising cell free therapeutic approach to control the infection and uncontrolled chronic inflammation in challenging diseases like PD. Thus, we may propose EXO, especially those derived from regDCs and enriched with immunoregulatory and/or antimicrobial factors, to be used for sustained local delivery and host immune modulation as an adjunctive to the conventional periodontal therapy. This could through injection of therapeutic EXO in the sites showing residual deep pockets that failed to respond to initial therapy. Moreover, the simultaneous use of EXO with SRP in periodontitis patients with incisor/molar patterns may provide additional immunological and antimicrobial benefits. However, the dose and duration of EXO-based therapeutics that may be proposed for periodontitis are still unclear. In addition, whether EXO should be employed for the prevention or resolution of periodontitis or both is another question that needs addressing. Moreover, early clinical trials are still necessary to evaluate the safety and efficacy of DCs EXO for periodontitis treatment.

## 5. Overall Conclusions

We conclude that conventional antimicrobial therapies for PD, such as antibiotics or antiseptics, are generally effective, in the short term, at reducing microbial load in the blood and tissues of PD patients, as well as improving clinical measures of PD. However, these approaches are non-selective and often too harsh for long-term use. Fortunately, under development are many more selective and targeted therapeutic approaches that indirectly act on microbial pathogens by preventing colonization, growth, and nutrient acquisition. Many of these approaches, traditionally called host modulation, have only become possible through decades of research on host-pathogen interactions and the immunopathogenesis of PD. More investment in such approaches is warranted and a goal worth pursuing, given the tremendous benefits that selective approaches might offer to patients with PD.

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
