# Peer review of "Selective Antimicrobial Therapies for Periodontitis: Win the “Battle and the War”"

_ijms, 2021, doi:10.3390/ijms22126459_

Round 1

Reviewer 1 Report

Finding a universal cure for periodontal disease is an unrealistic aspiration; clinicians must know that each case needs to be treated by ”patient-tailored” methods and adjunctive therapies represent an important tool. I think that the authors’ manuscript has the potential of providing valuable information for both clinical and scientific community, after addressing a few details:

  1. Please, provide reference for the following phrases:

”Systemic antimicrobials are principally indicated in immune-compromised PD patients or those with metabolic disorders that limit their response to mechanical therapy.”

”Subsequent clinical trials increased the dosing regimen to longer periods, showing reduction of inflammatory biomarkers. Cessation of therapy reversed collagenase activity to pre-treatment levels.”

”Plasma of mice with transmissible spongiform encephalopathy (TSE) contain EXO that harbor infectious misfolded prion proteins responsible for brain damage.”

”Serum EXO isolated from patients with tuberculosis infection were found to carry mycobacterial antigens.”

  1. Maybe, photodynamic therapy with various photosensitizers should also be taken into consideration.
  2. There is a slight inconsistency throughout the manuscript regarding the ”in-text” reference typing and the spacing between words.

Author Response

Comment. I think that the authors’ manuscript has the potential of providing valuable information for both clinical and scientific community, after addressing a few details:

  1. Please, provide reference for the following phrases:

”Systemic antimicrobials are principally indicated in immune-compromised PD patients or those with metabolic disorders that limit their response to mechanical therapy.”This has been added

”Subsequent clinical trials increased the dosing regimen to longer periods, showing reduction of inflammatory biomarkers. Cessation of therapy reversed collagenase activity to pre-treatment levels.” This has been added.

Plasma of mice with transmissible spongiform encephalopathy (TSE) contain EXO that harbor infectious misfolded prion proteins responsible for brain damage.”This has been added.

Serum EXO isolated from patients with tuberculosis infection were found to carry mycobacterial antigens.”This has been added.

Maybe, photodynamic therapy with various photosensitizers should also be taken into consideration.We thank the reviewer for this comment, and this has been added now in section 2.3.

There is a slight inconsistency throughout the manuscript regarding the” in-text” reference typing and the spacing between words.This has been addressed.

Reviewer 2 Report

This is a review article summarizing the attempts so far by clinicians and researchers regarding how adjunct periodontal therapy developed or should develop in the long run to help treated periodontitis patients and dentists to maintain health or prevent the high likelihood of chronic periodontitis recurrence. Special emphasis is on how best to modulate the host (including oral and/or systemic microbiota) so substained oral/periodontal and systemic health would be maintained, as well as highlights regarding how such field would/might develop in the future.

In general, the review is nicely written and provide great insights to IJMS readers interested in the topic/field. The Reviewer recommends acceptance and some minor revision:

  1. The Reviewer is concerned about over statement of the deleterious effects of antibiotics (section 2.1, last 2 sentences). The two cited references in fact indicated transient shift then bounce back (including GI flora) or only shifting of oral flora towards health upon successful treatment. Of course, dentists and physicians and alike should and need to exercise high level of care when prescribing antibiotics for patients with Clostridium difficile– positive ulcerative colitis history. Inclusive of such statements at the end of that paragraph would be considered “dramatizing the negativity of antibiotics adjunct periodontal therapy” – such unfair statements need to be modified or deleted. The same applies to the third sentence in the Conclusion section which is apparently not unequivocally supported by sound evidence especially the statement: “with potential for long term consequences to the health of the periodontal patient”. As a matter of fact, avocations of novel and patient/host/biological/environmentally friendly therapeutic approaches need not be built upon demeaning or patronizing an already existing therapy.
  2. The reference for probiotics (Section 3.1) seemed a bit old, perhaps the authors could consider citing more contemporary references say, within the past 5 years.Section 4 regarding EXO is exciting. Can the authors shed some lights regarding their anticipation of how EXO, (mDC, regDC, salivary, etc.) could be applied in modulation of the periodontitis host responses?Minor comment: 
  3. Page 2, first paragraph: ‘Gram-positive/negative’ should be ‘gram-negative’ (https://wwwnc.cdc.gov/eid/page/preferred-usage); same for pages 3 and 6.

Author Response

  1. The Reviewer is concerned about over statement of the deleterious effects of antibiotics (section 2.1, last 2 sentences).
    We agree with the reviewer and these statements have been either moderated or deleted.
  2. The reference for probiotics (Section 3.1) seemed a bit old, perhaps the authors could consider citing more contemporary references say, within the past 5 years. Section 4 regarding EXO is exciting. Can the authors shed some lights regarding their anticipation of how EXO, (mDC, regDC, salivary, etc.) could be applied in modulation of the periodontitis host responses?
    More recent studies regarding the adjunctive use of probiotics have been added now in section 3.1. The proposed application EXO in periodontitis was also added in section 4.2.
  3. Page 2, first paragraph: ‘Gram-positive/negative’ should be ‘gram-negative’ (https://wwwnc.cdc.gov/eid/page/preferred-usage); same for pages 3 and 6.
    This has been addressed.

Reviewer 3 Report

In this manuscript from Elashiry and colleagues reviewed the current and prospective antimicrobial therapies for PD. Unfortunately, the title is grand, but the content is simple and not new.

If the authors want to review about a ‘selective’ antimicrobial therapy for periodontitis, it is strictly needed to find the selective condition for antimicrobial therapy. Recently, several research and trials to realize a personalized medicine including multi-omics using next generation sequencing have been introduced. And local drug delivery system also has been developed for the treatment of periodontal disease. Based on these technological advances, it is needed authors find ‘selective strategy’ for the antimicrobial therapy. In addition, both sides need to be considered for a real victory in "Battle and War" because microorganisms and hosts cannot be considered separately.

In general, this review does not have depth, does not critically analyze the state of the art, furthermore, there is no figure or table to help readers understand. For these reasons, I think this manuscript is not suitable for IJMS.

Author Response

The point of this article, was exactly what the reviewer suggests, that is, microorganisms and hosts cannot be considered separately. We spend a lot of effort in the article reviewing traditional antimicrobial therapies, then explaining why host modulation therapy is antimicrobial therapy, moreover, why it is a more strategic approach than direct forms of antimicrobial therapy and what are the latest advances.  Based on the comments of this reviewer and the other reviewers we have extensively revised the article with additional advancements, reported the newest technologies, including complement C3 blocking peptides, etc.  The other two reviewers were very enthusiastic. I would love to see a review as the reviewer suggests, of personalized medicine including multi-omics using next generation sequencing, as applied to periodontitis research.